# Evaluation of Public Health Emergency Management in China: A Systematic Review

**DOI:** 10.3390/ijerph16183478

**Published:** 2019-09-18

**Authors:** Jia Wang, Beibei Yuan, Zhengmao Li, Zhifeng Wang

**Affiliations:** 1Department of Health Policy and Management, Peking University School of Public Health, 38 Xueyuan Rd, Haidian District, Beijing 100191, China; wangjiawst@163.com; 2China Center for Health Development Studies, Peking University, 38 Xueyuan Rd, Haidian District, Beijing 100191, China; beibeiyuan@bjmu.edu.cn; 3Health Emergency Response Office (Public Health Emergency Command Center). National Health Commission of the People’s Republic of China, No 1 Xizhimen Outer South Road, Xicheng District, Beijing 100044, China; lizm@nhfpc.gov.cn

**Keywords:** public health emergency management, evaluation system, systematic review, China

## Abstract

To summarize the present status of health emergency management assessment in China, a comprehensive search of Chinese databases for research that explicitly mention health emergency assessment indicators and indicator systems was performed. Studies were evaluated using the Ekman quality assessment tool, and data were extracted with an original extraction form. Sixty-one studies were included. There are many types and methods of health emergency management assessment in China, and the dimensions and the indicators involved are complex. Legal, regulatory, and policy bases for such assessment need to be further strengthened. The relevance of the entire assessment process and its practical application should be enhanced. In the occupational practice, appropriate evaluation methods should be selected according to respective evaluation purposes, evaluation objects, and contents. Laws, regulations, and policies in the evaluation of health emergency management should be improved. Finally, further correlational research on health emergency management evaluation system processes should be explored and improved.

## 1. Introduction

In 2003, a sudden outbreak of Severe Acute Respiratory Syndrome (SARS) occurred in China. More public health emergencies such as the H1N1 flu epidemic in 2009 and the H7N9 avian flu epidemic in 2013 had serious impacts on China’s politics, economy, trade, and people’s health, among others. Theory and implementation of health emergency systems have evolved worldwide. China has gradually implemented the construction of a comprehensive emergency management system including the use of theoretical models, assessment systems, and response studies. China’s health emergency management system is an important component of the comprehensive emergency management system, and health emergency management assessment is an important part of the health emergency management system. China’s health emergency management assessment system has experienced many rigorous tests in response to a series of public health emergencies and has accumulated experience in detecting health emergencies and managing the weaknesses of the evaluation system. There are many types of health emergency management assessments and many methods for health emergency management assessment in China, and the dimensions and the indicators of health emergency management assessment are complex. For example, the evaluation system named the “Disease control agency health emergency capacity evaluation index” has three dimensions, the system known as the “Evaluation index for public health emergency response capacity of township hospitals” has two dimensions, the system named the “National Health and Family Planning Commission, Health Assessment Capacity Assessment Standard” only has one dimension, the system called “A comprehensive assessment tool of the ability of all provinces, autonomous regions (municipalities)/prefectures (states) to respond to public health emergencies” has 214 indicators, and the evaluation system named the “Competency model of health emergency personnel in county-level disease control institutions” only has four indicators. In general, the dimensions of health emergency assessment are mostly second-level dimensions. The assessment indicators mainly include management capabilities, improvement capabilities, drill capabilities, and reporting capabilities. For purposes of this research, a “dimension” refers to the content of public health emergency management features of the institution, such as health emergency response dimension, and “indicator” is used to refer to evaluation features, including emergency command coordination mechanism. After the establishment of the National Emergency Management Department, it was necessary to pay attention to the differences and the connections between the health emergency and the national emergency laws and policies. It is necessary to improve the relevance of the entire process of the health emergency management assessment system. The practical application of the health emergency management assessment system also should focus on the application of health supervision agencies, schools, and other institutions. China includes a vast territory, a substantial population, and diverse regional cultures. In May 2005, the 58^th^ World Health Assembly (WHA) adopted International Health Regulations [2005] (hereinafter “IHR” or “the Regulations”), which were subsequently implemented on 15 June 2007. All state parties are required by the IHR to develop certain minimum core public health capacities. The development of China’s health emergency research has played an important role in promoting the development of global health emergency systems. Systematic evaluation as a method produces high-quality evidence, and its application in the collection and the production of scientific evidence in health emergency management assessment research serves to effectively guarantee the quality of research results, providing a scientific reference for researchers and policymakers. This study analyzes the characteristics of health emergency management assessment indicators using a qualitative system evaluation to appraise the current status of assessment in China as well as to provide evidence-based research for global health emergency policy development.

## 2. Materials and Methods

### 2.1. Document Inclusion and Exclusion Criteria

Build a model based on the Sample, Phenomenon of Interest, Design, Evaluation, Research Type (SPIDER) qualitative system evaluation questions to hypothesize about problems and develop inclusion exclusion criteria.

### 2.2. Inclusion Criteria

Research object (sample): China Health Emergency Assessment Research.Research content (phenomenon of interest): explicitly mention health emergency assessment indicators and the health emergency assessment indicator system.Incorporate the research design (design): randomized controlled trial (RCT), non-randomized controlled trial (non-RCT), controlled trial (CBA), time series (interrupted) studies, and cross-sectional studies.Evaluation: the basic characteristics of China’s health emergency assessment indicator system and the main indictors according to laws and regulations and their application.Research Type: qualitative or quantitative research.

### 2.3. Exclusion Criteria

Repeat published studies.Yearbooks, patents, conference abstracts, personal comments, letters, newspaper articles, work plans, and summaries.Theoretical research that only mentions the word “indicators of health emergency assessment” but has no specific evaluation index content or only partial evaluation indicators, epidemiological characteristic analysis, and epidemiological monitoring reports.There is no valid content literature, or the original language is from non-Chinese literature and the literature cannot be downloaded.Evaluation results fall within the scope of low quality research after using the Ekman quality assessment tool [1].

### 2.4. Literature Search Strategy

A comprehensive system search of Chinese databases—including the China Knowledge Network (CNKI), the Wanfang Data Knowledge Service Platform (WANFANG DATA), and the VIP (VIP data platform)—was implemented that used a search time limit from the establishment of the database as of March 2019. According to the type, the scope, and the research purpose of health emergency management combined with a preliminary search of China’s domestic health emergency management assessment literature, two parts of search strategies were identified. The first part included literature encompassing health emergency, public health emergencies, and health and epidemic prevention, and the second part included evaluation, assessment, monitoring, and performance. Each part of the search terms were connected by the logical symbol “OR”, and two parts were connected by the logical symbol “AND” to obtain the search results. According to this principle, the corresponding search formulas were formulated according to the characteristics and the requirements of each database. The specific search strategy is detailed in Table 1.

### 2.5. Literature Screening

The literature screening was conducted in two phases. In the first stage, all search documents were found in the form of bibliographies and then exported to the database for review. The searched documents were checked by topic and abstract. In the second stage, the full text was found and read to determine if it should be included. In the above stage, two evaluators independently screened the literature. If any disagreement arose, experts in the field of health emergency and evidence-based medicine were consulted to assist with judgment. The above document screening process was based on NoteExpress.

### 2.6. Data Extraction

A self-made data extraction table was used to extract the data. The data extraction content mainly included: (1) basic characteristics of research (i.e., research topic, first author, publication time, region, research field, article type, evaluation content, research institution or research objects, research methods, and main research processes); (2) research evaluation system dimensional characteristics (i.e., name of research evaluation system, classification of research institutions, number of dimensions, and the composition of indicator system); (3) study indicators (i.e., first-level indicator and most frequently used indicator); (4) application of research evaluation system; (5) legal, regulatory, and policy on the basis of research evaluation system [i.e., evaluation system evaluation content, evaluation system, construction laws and regulations, and the system (or other standards)]. The data extraction process relied on NVivo12.

### 2.7. Quality Evaluation Criteria

The two reviewers independently screened the literature and conducted a literature quality evaluation. If there were any inconsistencies, they were discussed and resolved first. If any differences persisted, an expert was consulted to assist in the judgment. Ekman’s quality evaluation list was used to evaluate the quality of the included studies. The evaluation includes 7 aspects, i.e., research analysis, rationality, methodology, data, goal achievement, results, discussion, and conclusions. Each evaluation indicator corresponds to a score; three “*”s indicate high quality research, two “*”s indicate medium quality research, and one “*” indicates low quality research performed.

### 2.8. Data Analysis

Based on the characteristics of the included studies and the extracted data, qualitative data were used to analyze the collected research data.

## 3. Results

### 3.1. Literature Screening

According to the search strategy, a total of 9657 articles were obtained in the database, including 2201 CNKI articles, 6218 WANFANG DATA articles, and 1238 VIP articles. Thereafter, 2733 duplicate articles were excluded in NoteExpress, 6924 articles remained, and four references were tracked. According to the inclusion and the exclusion criteria, the reading topics and the abstracts were screened, and 6705 unrelated research documents were excluded, leaving only 223 that were screened. The retrieval process is depicted in Figure 1.

### 3.2. Basic Characteristics

All 61 studies (included) were cross-sectional studies with 36 studies from East China (59%), 12 from Central China (19.7%), and 13 from West China (21.3%). In terms of research literature types, 38 studies (62.3%) were journal type documents, and 23 studies (37.7%) were dissertation type documents [seven doctoral theses (30.4%) and 16 masters’ theses (69.6%)]. In the research field, there were 39 studies on health emergency response capability (63.9%), one health emergency manpower resource allocation efficiency evaluation study (1.6%—[2]), one vulnerability public health event vulnerability evaluation study (1.6%—[3]), three drill evaluation studies (4.9%—[4,5,6]), four comprehensive treatment evaluation studies (6.6%—[7,8,9,10]), three plan evaluation studies (4.9%—[11,12,13]), one quality of management teaching case evaluation study (1.6%—[14]), two related department setting and system construction evaluation studies (3.2%—[15,16]), two monitoring and early warning evaluation studies (3.2%—[17,18]), one epidemiological characteristics and normative disposal of third-party evaluation study (1.6%—[19]), one social vulnerability assessment study (1.6%—[20]), one emergency performance appraisal and evaluation study (1.6%—[21]), one social mobilization mechanism study (1.6%—[22]), one quality evaluation of network direct report (1.6%—[23]), and one responders’ competency evaluation study (1.6%—[24]). The representative research of China’s health emergency assessment system is mainly distributed in the eastern part of China. The most common type of publication is papers in academic journals, and the main type of research is on health emergency response ability. The research design is primarily cross-sectional. According to the classification of health emergency management, subjects most frequently include research on health emergency management evaluation of health administrative departments, disease prevention and control institutions, medical and health institutions, health supervision institutes, schools, military, and ports.

### 3.3. Quality Evaluation

The quality of the preliminary inclusion studies was assessed using the Ekman quality assessment tool. In terms of methodology, 60 studies (98.4%) clearly described the health emergency assessment research methods, of which 22 studies clearly indicated the time period of the cross-sectional study. In terms of data, 60 studies’ data (98.4%) were derived from original data, while only one study (1.6%) used secondary data. The scores indicated that three studies (4.9%) were high quality, and the remaining 58 studies’ data (95.1%) were categorized as medium quality research. Overall, most studies were given a rating of 2* according to the Ekman quality assessment tool, which scores between 17 and 21; the research quality of China’s health emergency management assessment system is relatively good. The specific results are detailed in Table 2, and the numbers 1–61 represent the research serial number.

### 3.4. Dimensional Characteristics of Assessment System

Fifty-five studies have established health emergency assessment systems. Six studies were based on China’s Health Emergency Response Survey and Evaluation Standards, China’s Public Health Emergency Report Management Information System, and China’s former Ministry of Health’s disease prevention and control performance appraisal operation. The manual (2009 Edition) and the Chinese Health Department IHR and other standards conducted a comprehensive health emergency assessment. The health emergency assessment system in 24 studies consisted of three dimensions, such as monitoring capability, plan capability, management capabilities, or similar constructs. Overall, in the health emergency assessment systems, the maximum and the minimum number of indicators were as follows: all indicators (maximum of 214, minimum of 4); primary indicators (maximum of 15, minimum of 2); secondary indicators (maximum of 204, minimum of 0); tertiary indicators (maximum of 84, minimum of 0). The specific results are detailed in Table 3.

### 3.5. Indicator Characteristics and Application of Assessment System

The indicator characteristics and the application of the health emergency assessment system included in the study were analyzed according to different institutional categories, which were “State and Health Administrative Department”, “Disease Prevention and Control Institutions”, “Health Institution”, “Health Supervision Agency”, “School”, “Military”, and “Border Port Health and Quarantine Department”.

#### 3.5.1. State and Health Administrative Department Assessment Research

Twenty-three studies were conducted on health emergency assessment at all levels of government and health administrative departments. The indicators that were most frequently used were management capabilities (1.45%, including control capabilities, command capabilities, and organization capabilities) and response capabilities, improvement capacity (2.49%, including recovery, education, reconstruction, and preparation capacity), drill capability (1.99%), reporting capacity (2.24%), response system (and system construction) (2.24%), monitoring capability (1.99%), materials reserve capacity (1.74%), training capability (1.74%), early warning capability (1.74%), social mobilization ability (1.49%), site disposal capacity (1.49), and risk assessment (1.24%).

#### 3.5.2. Disease Prevention and Control Institutions Assessment Research

Twenty studies were concerned with the health emergency assessment of disease prevention and control institutions, and the indicators that were most frequently used were technical expertise (8.31%), management capabilities (2.56%, including treatment, control, operation, and organization), acquired ability (2.26%), participation ability (2.72%), response capacity (3.38%), monitoring capability (3.38%), early warning capability (2.77%), response capability (2.46%), drill capacity (2.15), materials reserve support capacity (5.55%), training capacity (1.85), system construction (1.54%), cooperation capacity (1.23%), and team building capacity (1.23%).

#### 3.5.3. Health Institution Assessment Research

Nine studies were concerned with health emergency assessment in health institutions. The indicators most frequently used were health emergency monitoring (4.82%), plan (4.22%), material reserve (3.61%), training (3.01%), early warning (3.01%), management command (2.41%), information monitoring report (2.41%), institutional system (2.41%), personnel (2.41%), medical technology (2.41%), education (1.81%), laboratory (1.81%), medical treatment (1.81%), bed (1.20%), logistics support (1.20%), communication (1.20%), on-site disposal (1.20%), and drills (1.20%).

#### 3.5.4. Health Supervision Agency Assessment Research

Two articles were concerned with the health emergency assessment of health supervision agencies. The indicators most frequently used were organization command (10.10%), management system (5.05%), equipment reserve (12.12%), training drill (12.12%), and emergency response (6.06%).

#### 3.5.5. School Assessment Research

Two articles were concerned with school health emergency assessment research. The indicators most frequently used were health emergency value mission (13.79%), preparation and recovery (6.9%), detection and monitoring (3.45%), response (3.45%), and materials learning (3.45%).

#### 3.5.6. Military Assessment Research

Five articles were concerned with military health emergency assessment research. The indicators used most frequently were health emergency command capacity (9.72%, including organizational capacity and classification ability), support capability (4.17%), equipment (5.56%), system and system construction (2.78%), response capacity (2.78%), monitoring capability (2.78%), medical treatment capacity (5.56%), early warning capability (2.78%), education (1.39%), control capacity (1.39%), service capacity (1.39%), survivability (1.39%), team construction (1.39%), response capability (2.78%), research capacity (1.39%), drill (1.39%), rescue capability (1.39%), education (1.39%), and drug reserve (1.39%).

#### 3.5.7. Border Port Health and Quarantine Department Assessment Research

Two studies conducted health emergency assessment research on the health and quarantine departments at border ports. The most frequent indicators used were health emergency vulnerability (9.68%), environment (6.45%), system construction (6.45%), laboratory construction (3.23%), network construction (3.23%), drill (3.23%), support (3.23%), on-site disposal measures (9.69%), monitoring (3.23%), training (3.23%), and warning (3.23%). 

#### 3.5.8. Health Emergency Assessment Practice Research

Health emergency assessments conducted at all levels of government and health administrations are mainly used in Shanghai, Sichuan, China, and Guangdong Provinces. Health emergency assessments conducted at disease prevention agencies are mainly used in Shanghai, Shandong, Guangxi Zhuang Autonomous Region, Guangdong, Inner Mongolia Autonomous Region, the entire country, and Henan Province. Health emergency assessment in medical and health institutions is mainly applied in Guangxi Zhuang Autonomous Region, Hebei Province, Heilongjiang Province, the entire country, and Beijing. Health emergency assessment in the military is mainly applied to the entire army. Health emergency assessment conducted at the port is mainly applied to Xinjiang’s border ports. 

Through this research, we gained a comprehensive understanding of the status quo of China’s health emergency assessment and explored the main distribution areas of the representative research of China’s health emergency assessment system, research output, research design, main dimensions, high frequency indicators, major application cities, and health emergencies. We further evaluated the subjects of the classification. The study only included Chinese studies, which affects the comprehensiveness of the research. The specific results are shown in Table 4.

### 3.6. Legal and Policy Basis for the Evaluation System

Twenty-seven studies (44.3%) referred to laws, regulations, systems, or other standards during the emergency assessment system construction process. Among them, 18 studies (66.7%) were evaluated for health emergency response, one study (3.7%) on a social vulnerability assessment system for major infectious diseases, one study (3.7%) on an early warning system for influenza outbreaks, two studies (7.4%) on health emergency pre-plan assessments, one study (3.7%) on performance assessment, one study (3.7%) on teaching case assessments (on emergency management), and three studies (11.1%) on health emergency work evaluation.

## 4. Discussion

Health emergency management assessment is an indispensable part of health emergency management. Through proper evaluation, problems and deficiencies in health emergency management can be detected over time to ensure and enhance the efficacy by which such measures can be implemented in time to maximize the likelihood of preventing and controlling public health emergencies.

### 4.1. Wide Range of Assessments

In health emergency management assessment practice, the types of assessments vary according to different classification criteria such as the purpose, the object, and the content of the assessment as well as the work status [10]. Research clearly indicates that there are a multitude of assessments in China. First, assessments are divided according to purpose, i.e., formative and summative assessments. For example, the evaluation of Yunnan Province’s emergency health response capability and core competence assessment index system at district-level control institutions was performed to guide and advance the progress of assessment targets as well as to provide a formative evaluation of management decisions. On the other hand, in Fujian Province, the study is concerned with the state of performance appraisals regarding public health emergencies and other issues, including formulating (such as plan completion rate, emergency material reserve rate, simulation exercises) overall judgments on the purposes of evaluation institutions, including accountability and summative assessment. Second, health emergency management assessment is divided into several stages, including the preparation stage (i.e., pre-emergency evaluation), the disposal stage, the in-process evaluation, and the recovery stage (i.e., post-emergency evaluation). Thus, a preliminary discussion on the early warning index system for influenza outbreaks, the construction of an evaluation index system for hospital nurses’ public health emergency response capacity, a weighted analysis of emergency public health emergency response evaluation factors, and the evaluation of emergency drill activities in county-level disease control institutions in Nanchang City are all examples of pre-assessment. In contrast, social vulnerability assessments and analyses of significant factors associated with major infectious diseases are concerned with the assessment of the nature, the type, the extent, and the determining factors of public health emergencies, while epidemiological characteristics studies and normative disposal third-party assessments are all considered post-event assessments that focus on recovery, summarizing, and long-term impact. Third, emergency management work assessment can be divided according to health status, i.e., normal and abnormal assessment. For example, the analysis and the evaluation of the present state of affairs of health emergency work in Qinghai Province was a normal assessment of the state of daily management construction, whereas the comprehensive evaluation of public health emergency implementation in the Fujian Province in 2014 was an assessment of the process and the impact of the emergency response. Fourth, assessment is divided according to the evaluation implementation body, i.e., internal and external evaluation. For example, the evaluation of the public health emergency response capacity of F hospital in Qiqihar City was an internal investigation conducted by an investigation team composed of health emergency management agencies and workers. On the other hand, investigating the epidemiological characteristics of public health emergencies and standard treatment third-party assessments are surveys that involve external evaluation from experts outside the health emergency management and work organization.

### 4.2. Health Emergency Management Assessment Methods

Health emergency management assessment is a process of formulating objective judgments on assessment targets based on specific criteria. Qualitative, quantitative, and mixed methods are most commonly used. This review found that the construction of a health emergency management evaluation system is more of a qualitative approach and is appropriately categorized as exploratory research (e.g., literature review, personal interview, Delphi, brainstorming, expert group discussion, expert meeting business law, field inspection). Some studies focus on data collection, detailing the main laws and characteristics by using descriptive methods. The health emergency management evaluation system weight analysis and application falls more in the domain of the Delphi method, while analytic hierarchy process, entropy weight, rank sum ratio, fuzzy comprehensive evaluation, and matter-element extension methods are relatively new methods for health emergency assessment. The application of the health emergency management evaluation involves possibly utilizing questionnaire surveys, factor analysis, principal component analysis, comprehensive scoring, comprehensive indexes, and analytic hierarchy process methods. Recently, scholars have started to pay attention to health emergency resource allocation research using data envelopment analysis methods. Health emergency management systems are extremely complex and large, as they are concerned with numerous variables and complicated mechanisms and structures. 

### 4.3. Health Emergency Management Assessment Dimensions and Indicators Are Complex

The dimensions and the indicators of health emergency management assessment represent, to some extent, a specific aspect of health emergency management work and represent the constituent elements of health emergency management (e.g., health emergency agencies, manpower, materials, information technology, funds). These elements support health emergency management work and are the foundation for the smooth implementation of health emergency management. This review found that the assessment dimensions predominantly included national and local state health administrative departments’ normal emergency management and abnormal emergency command organization management. Assessment dimensions were also concerned with health emergency professional technical institutions, including health emergency level disease prevention and control institutions, professional disposal management, personnel management, materials management, information management, fund management, etc., at various levels of health institutions. Moreover, health emergency assessment indicators are embodied in the duties and the authority of the state in the management of normal/abnormal emergencies, duties of professional technical institutions, classifications and responsibilities of technical and health emergency personnel, emergency materials categories and management methods, information monitoring, source, report, confidentiality, and supervision, and aspects of health emergency fund sources to support policy and supervision.

### 4.4. Legal and Policy Basis Require Further Reinforcement

Due to the differences in organization, system, economy, culture, and history of various countries, the organization, the function, and the structure of their respective national health emergency management and evaluation systems differ. In April 2018, the central and the state institutions of China reformed and established the National Emergency Management Department, equipped with primary functions responsible for the management of natural and accidental disasters. Social security incidents were assigned to the Political and Legal Committee, and public health events became the responsibility of the Health Committee. In the future, more comprehensive, targeted, and operational public health events, social security incidents, natural disasters, and accident disasters related laws and regulations and various policy systems will likely be gradually developed, and corresponding event-related authorities will also enhance the supporting plan, the disposal plan, and the process. This study found that the main Chinese environmental laws and regulations utilized in the process of constructing a health emergency assessment system included the Emergency Regulations for Public Health Emergencies, the Law of the People’s Republic of China on Emergency Response, the Law of the People’s Republic of China on the Prevention and Control of Infectious Diseases, “The disease prevention and control work performance appraisal operation manual”, the State Council Emergency Office “Emergency Drill Guide”, and so on. In addition, the International Health Regulations (IHR2005), the “US Center for Disease Control (CDC)” public health preparation, and the response capacity scale “Expert Advisory Standards”, in addition to others, were also used. Nonetheless, the research in reference to laws, regulations, systems, or other standards is still marginal, and there persists a lack of detailed studies on legal, regulatory, and policy bases for health emergency management assessment before the establishment of the National Emergency Management Department. After the establishment of the National Emergency Management Department in April 2018, China’s emergency management functions were gradually clarified. Laws, regulations, and policies for health emergency management assessment need to be continuously updated, and in-depth research needs to be conducted in order to provide a basis for health emergency management assessment. See the Appendix A.

### 4.5. Relevance of Health Emergency Management Evaluation System 

The various system elements of the health emergency management system are interrelated and are organic as a whole, and the connecting paths between various elements are worth further exploration. This study found that most research is narrowly focused on particular topics; indeed, evaluation research is typically carried out on specific problems. Yet, it is rare to explore the interrelationships and the causal paths among the various elements of the evaluation system. Specific problems should be found in the health system, and relationships and mechanisms of system elements should undergo in-depth analysis to accurately determine the underlying causes of a health emergency problem and fully comprehend the full process management of public health emergencies. 

### 4.6. Need to Strengthen Practical Application of Assessment Systems 

This study found that research on constructing health emergency assessment systems, analyzing and evaluating present systems based on existing standards, and evaluating practical applications based on the construction of management evaluation systems is scarce. In the application evaluation research, the types of application institutions are predominantly from disease prevention and control institutions, medical and health institutions, government and health administrative departments, and military and border ports. Merely building a health emergency assessment system cannot meet the needs to form a comprehensive emergency system, nor can it identify the weaknesses in management efficiently. Practical research on the evaluation of health emergency systems should be conducted rigorously, and attention should be paid to the actual evaluation and application of health supervision institutions, schools, and other institutions.

### 4.7. Limitations and Strengths

The study solely included Chinese research, and this arguably limits the ecological validity of the findings. Yet, this study still conducted a broad and comprehensive investigation of the Chinese literature, traced references, and conducted an in-depth analysis of medium-quality and high-quality research, effectively reducing bias due to insufficient literature included.

## 5. Conclusions

The qualitative system evaluation of China’s health emergency management assessment revealed that, although China has carried out substantial research related to health emergency assessment, in the face of frequent global public health emergencies, the current assessment of health emergency areas still requires continuous improvement. Foreign health emergency management assessment related research is carried out earlier and applied in practice. The research on China’s health emergency management assessment has been carried out late, the evaluation system is flawed, the evaluation criteria are not standardized, the evaluation model is unstable, and the assessment of health emergency management processes lacks attention. A first step in improving the situation is to enhance the relevance of the process of health emergency management assessment. For more than ten years, the health emergency management assessment has prioritized the evaluation of staged and key management work. At present, there is still a lack of research across all processes related to the health emergency management evaluation system. Second, the process of the health emergency management evaluation system is weak. Previous research on a health emergency management assessment system typically targeted certain types of public health emergencies and lacked the exploration of health emergency management assessment systems based on the commonality of multiple types of public health emergencies. Third, the application of a health emergency management assessment system is not strong. The content and the objectives of the previous health emergency management assessment systems are relatively scattered, lack organic integration of the assessment process, and create a state of under-applied appraisal.

The representative research of China’s health emergency assessment system is mainly distributed in the eastern part of China. The design of such research mainly focuses on cross-sectional studies. While the dimensions and the high-frequency indicators of the health emergency assessment system have been extracted, the applicability of the health emergency assessment system needs to be improved. China should emphasize research and application of health emergency assessment in central and western regions, including ethnic minority areas, and the health emergency assessment system should be improved from the perspective of classification and integration of health emergency assessment subjects, as health emergency management assessment is an important part of the health emergency management activities. Through evaluation, problems in health emergency management can eventually be found, and the effective prevention and control of public health emergencies can be achieved to the greatest extent possible. There are many types of and methods for health emergency management assessments, and the related dimensions and indicators are complex. In health emergency management assessment, it is extremely important to select appropriate assessment methods according to different assessment purposes, targets, and contents. Laws, regulations, and policy bases of health emergency management assessment should be strengthened, especially after the establishment of the National Emergency Management Department in April 2018. 

The most important finding of the current research is that there are many types of health emergency management assessments and multiple health emergency management assessment methods in China. The dimensions and the indicators of health emergency management assessment are also complicated. In general, the dimensions of health emergency assessment are mostly in the secondary dimension. The evaluation indicators mainly include management ability, improvement ability, exercise ability, and reporting ability. Laws, regulations and policy bases for health emergency management assessment also need to attend to differences and linkages between national health emergencies and national emergency laws, regulations, and policies. It is necessary to improve the relevance of the entire process of the health emergency management assessment system. The practical application of the health emergency management assessment system also needs to pay attention to the application of health supervision, schools, and other institutions. The correlation research of the entire process of the health emergency management evaluation system needs to be further explored and improved. In addition, China should build a health emergency management assessment system that is in line with international standards and focuses on Chinese characteristics, which needs further exploration in the future.

China’s comprehensive promotion of health infrastructure construction requires improvement of disaster prevention, mitigation/emergency response capabilities, and improvement of the emergency health emergency response system. Future research should explore the optimal methods of realizing China’s health emergency management, evaluate relevance and commonality, and develop sound and applicable standardization of health emergency management assessment systems to meet the developing requirements of potential emergencies in China. This research can be used to develop health emergency assessment research in other countries and contribute to the development of a global health emergency movement.

## Figures and Tables

**Figure 1 ijerph-16-03478-f001:**
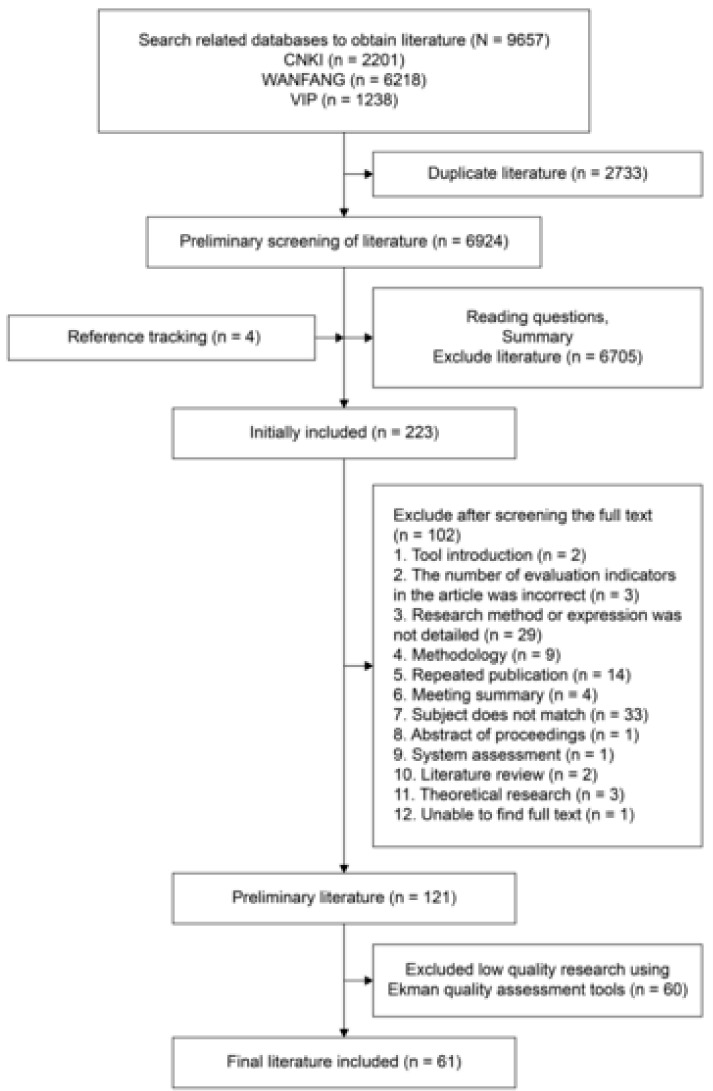
Flow chart of study selection process.

**Table 1 ijerph-16-03478-t001:** Search terms and search strategies.

Database	Literature Search Strategy
CNKI	
#1	SU = “health emergency”
#2	SU = “public health emergency”
#3	SU = “health and epidemic prevention”
#4	SU = “assessment”
#5	SU = “evaluation”
#6	SU = “monitor”
#7	SU = “performance”
#8	(#1 OR #2 OR #3)AND(#4 OR #5 OR #6 OR #7)
WANFANG	
#1	theme:(health emergency)
#2	theme:(public health emergency)
#3	theme:(health and epidemic prevention)
#4	theme:(assessment)
#5	theme:(evaluation)
#6	theme:(monitor)
#7	theme:(performance)
#8	(#1 OR #2 OR #3)AND(#4 OR #5 OR #6 OR #7)
VIP	
#1	M = health emergency
#2	M = public health emergency
#3	M= health and epidemic prevention
#4	M = assessment
#5	M = evaluation
#6	M = monitor
#7	M = performance
#8	(#1 OR #2 OR #3)AND(#4 OR #5 OR #6 OR #7)

CNKI: China Knowledge Network; WANFANG: Wanfang Data Knowledge Service Platform.

**Table 2 ijerph-16-03478-t002:** Ekman quality evaluation scores of studies.

Ekman Quality Evaluation Score	Study
3: 16–0	None
2: 21–17	1 [25], 2 [26], 3 [27], 4 [2], 5 [28], 6 [29], 7 [30], 8 [31], 9 [32], 10 [7], 11 [3], 12 [33], 13 [34], 14 [4], 15 [35], 16 [8], 17 [36], 18 [11], 19 [37], 20 [38], 21 [14], 22 [15], 23 [5], 24 [39], 25 [17], 26 [12], 27 [13], 28 [6], 29 [19], 30 [9], 31 [20], 32 [10], 33 [40], 34 [41], 35 [42], 36 [43], 37 [44], 38 [16], 39 [45], 41 [21], 42 [46], 43 [47], 44 [48], 45 [49], 47 [22], 48 [50], 49 [23], 50 [51], 51 [52], 52 [53], 54 [54], 55 [55], 56 [56], 57 [57], 58 [58], 59 [24], 60 [59], 61 [60]
1: 25–22	40 [18], 46 [22], 53 [61]

**Table 3 ijerph-16-03478-t003:** Dimensional characteristics of evaluation system studies.

No	Indicator System Name (Build System/Application Existing System)	Number of Dimensions	Number of Indicators	Composition of Primary Indicators
1	Disease control agency health emergency capacity evaluation index.	1	11	number of drills, number of participants, number of teams, number of senior titles in team, number of trainings, training participants, number of vehicles, protective materials, sanitation disposal state, special funds, communication equipment.
2	Evaluation index for public health emergency response capacity of township hospitals.	2	15	planning, monitoring, early warning, and service capabilities.
3	Disease control agency health emergency capacity assessment index.	3	115	system construction, team building, monitoring and early warning capabilities, response capabilities, support capabilities, and information communication and departmental collaboration, research, and cooperation and communication capabilities.
4	Public health emergency preparedness assessment index for provincial Center for Disease Control (CDC).	2	59	emergency management and coordination, risk monitoring and information management, reserve materials, site disposal, detection, and safety protection.
5	Human resource allocation efficiency evaluation index.	3	24	number of employees in the organization, proportion of employees with undergraduate degrees or greater, and emergency knowledge training participation rate in past three years.
6	Quantitative evaluation of overall school response capabilities to public health emergencies.	3	49	pre-preparation, event discovery, event handling, and post-recovery abilities.
7	Evaluation index of rural grassroots emergency response capability to public health emergencies.	3	74	human, financial, material, information, and technical resources, and population health levels.
8	Comprehensive evaluation of emergency response capabilities of disease prevention and (disaster) control institutions in Zhejiang (county).	3	106	system construction, team construction, support, response, monitoring, and early warning, scientific research and cooperation and communication, information communication and departmental collaboration capabilities.
9	Comprehensive evaluation model of county general hospitals response capabilities to public health emergencies.	2	37	command coordination, emergency plan, monitoring and early warning capability, information report exchange, emergency personnel, emergency beds, and material reserve.
	Comprehensive evaluation model of township hospitals response capabilities to public health emergencies.	2	15	emergency documents, emergency monitoring, and capacity building.
10	Guangxi CDC Public Health Emergency Response Capability Evaluation Index.	3	79	emergency organization system, emergency team construction, monitoring and early warning capability, actual emergency response capability, emergency protection, information communication, and departmental collaboration capabilities.
11	Shanghai Urban Health Emergency Core Competency Evaluation Index.	2	41	organizational command, work norms, emergency teams, equipment reserves, training drills, monitoring and early warning capability, publicity and education, laboratory energy, and force and emergency response.
12	Multi-criteria crisis early warning extension model (based on Public Health Emergency of International Concern (PHEIC) subject vulnerability assessment).	2	16	disaster response, population, environmental, and safety measure risk factors.
13	Guangxi county-level health and family planning bureau response capacity for public health emergencies.	2	30	emergency management and command, monitoring and early warning capability, information reporting, on-site disposal and personnel, material reserves and exercises, and recovery and evaluation.
14	National Health and Family Planning Commission, Health Assessment Capacity Assessment Standard.	1	8	system construction, emergency team, equipment reserves, training drills, mission research, monitoring and early warning capability, emergency response, and after-care assessment.
15	Self-made drill evaluation index.	2	27	pre-rescue preparation, epidemic situation verification and consultation, on-site investigation, laboratory testing, epidemic control, risk communication and health education, and emergency termination and summary report.
16	National Health Emergency Response Survey and Evaluation Standards.	1	8	system construction, emergency team, equipment reserve, training drill, mission research, monitoring and early warning capability, emergency response, and aftercare assessment
17	Public Health Emergency Report Management Information System.	1	7	number of public health emergencies, county reporting rate, monitoring sensitivity, timeliness of reporting, control effects, integrity rate, and accuracy.
18	Comprehensive evaluation index of hospital nurses’ ability to respond to public health emergencies.	2	24	basic information, professional background, knowledge system, and practical skills.
19	Tertiary hospitals’ emergency rescue plan effectiveness evaluation index.	2	21	plan integrity, operability, efficiency, flexibility, sociality, and plan management.
20	Public Health Emergency Work Ability Questionnaire.	2	25	work capacity self-evaluation, professional knowledge training effect, work capacity constraints, factors improvement, and work ability prediction.
21	Construction of an evaluation system and development of software for the evaluation index of the capacity of the armed police unit’s health emergency rescue team.	3	65	including organizational command, emergency maneuver, injury and treatment, medicinal materials protection, classified and sent, epidemic prevention and anti-health preservation.
22	Indicator system for health emergency teaching case evaluation.	3	47	including material value dimension, structure value dimension, practical value dimension and literary value dimension.
23	Community Medical Institutions Health Emergency Capability Survey and Evaluation Form in Guangdong Province.	2	34	condition of community medical institutions, community health human resources, health emergency related personnel and department setting, health emergency related work system, and health emergency plan construction.
24	2013 National Nuclear and Radiation Emergency Health Emergency Team Exercise Evaluation System.	2	35	program development, exercise preparation, on-site drills, and summary assessment.
25	Emergency response capability evaluation index system for disease control institutions in Guangdong Province.	3	92	emergency management system construction, emergency human resources, monitoring and early warning capabilities, emergency response capabilities, laboratory testing capabilities, emergency support capabilities, training and drills, and health education and media communication.
26	Tertiary monitoring and early warning system framework.	3	62	including monitoring system, risk assessment system, early warning system, and system guarantee.
27	Index system for evaluation of community health emergency plan.	3	63	structure, process, and results.
28	Emergency plan for public health emergencies.	1	7	time factors, personnel loss factors, economic loss factors, social impact factors, resource consumption factors, transportation and security factors, and program dynamic adjustable factors.
29	Evaluation of emergency drill activities of county-level disease control institutions in Nanchang City.	1	4	program development, exercise preparation, on-site drills, and summary assessment.
30	Disease Prevention and Control Work Performance Assessment Operation Manual (2009 Edition).	1	6	event report, event confirmation, event preparation, event site disposal, control measures implementation, and summary assessment.
31	Emergency evaluation index system for public health emergencies.	3	72	preparation stage evaluation, monitoring and early warning stage evaluation, response process evaluation, and post-event evaluation.
32	Constructing a social vulnerability assessment function for sudden epidemics of major infectious diseases.	2	14	social system vulnerability and social system resilience.
33	Comprehensive evaluation index system for the impact of public health emergencies.	3	75	health effects, economic impact, and social impact.
34	Hospital emergency ability evaluation index system.	3	56	emergency system, emergency agencies, monitoring and early warning of public health emergencies, on-site rescue and medical treatment, logistics support, emergency training and drills, and public education.
35	Comprehensive Evaluation Model of Xinjiang Frontier Ports’ Emergency Response Capability for Major Infectious Diseases.	3	101	basic conditions, emergency response system, monitoring and early warning capability, emergency support, laboratory capabilities, on-site disposal capabilities, information network systems, training, and exercises.
36	Disease Prevention and Control Center Infectious Disease Prevention and Control Capability Evaluation Index System.	2	61	comprehensive guarantees, immunization prevention, infectious disease emergency plans and drills, infectious disease monitoring, on-site disposal capabilities, information analysis and utilization, laboratory capabilities, publicity and education, and training.
37	Grassroots preventive health care centers emergency response capacity evaluation index.	2	29	organization management, technology implementation, resource reservation, monitoring and early warning, coordination, and cooperation.
38	County-level disease prevention and control institution emergency capability evaluation.	2	15	emergency preparedness, monitoring and reporting, emergency response, after-treatment, and integration.
39	Main indicators reflecting the emergency response capacity of health institutions at all levels.	1	15	number of institutional staff, emergency team count, number of emergency team members, number of senior members among emergency team members, simulation exercises, number of times, number of training courses in unit, number of participants in training class, number of participants in emergency training, number of emergency special vehicles, number of emergency on-site inspection vehicles, value of physical reserves, daily work expenses of the emergency department, annual emergency budget reserve, total number of beds (health institutions), and number of emergency beds.
40	County-level CDC emergency public health emergency response capability evaluation index system.	3	31	resource allocation, capacity building, and function implementation.
41	Influenza outbreak early warning indicator system suitable for China’s national conditions.	3	36	pre-emergency, atypical symptoms, and typical symptom.
42	According to the Ministry of Health, December 2008, the basics of disease prevention and control institutions at all levels. Responsibilities and Disease Surveillance Control Performance Evaluation Standards.	1	5	completion rate of the plan system, simulation exercise index, reserve rate of emergency items, standard disposal index, and event investigation rate.
43	Digital hospital emergency public health response capacity evaluation index.	3	82	emergency system, monitoring and early warning, medical treatment inside and outside the hospital, emergency reserve, personnel and equipment safety, education, and improvement.
44	Township public health emergency response evaluation system.	3	72	regional socio-economic population status, public health emergency basic support capacity, disease prevention and control, and health emergency business development level.
45	Tianjin City and County Disease Control Agency Emergency Capability Priority Improvement Indicators.	2	27	organizational management, material resources, information management, and professional skills.
46	Assessment of suitability of China’s existing emergency response capability system evaluation framework and index system.	2	37	primary disease prevention and control center: plans, monitoring, laboratories, manpower, information, training, disposal, and reserves.
		2	36	county-level general hospitals: documents, monitoring, laboratories, manpower, information, training, emergency, and reserves.
		2	15	township hospitals: documentation, monitoring, and capacity building.
47	Comprehensive evaluation index of social mobilization apparatus for public health emergencies.	3	59	domestic unified command, mobilization of other social resources, human mobilization, information culture mobilization, material mobilization, and economic mobilization.
48	Constructing hospital emergency response evaluation index.	3	56	emergency system, emergency agencies, monitoring and early warning of public health emergencies, on-site rescue and medical treatment, logistics support, emergency training and drills, public awareness, and education.
49	Evaluation of emergency capability evaluation system.	2	29	including forecasting and early warning capabilities, technology implementation capabilities, resource reserve capacity, operational management capabilities, and access to foreign aid.
50	Network Quality Evaluation Index for Public Health Reports.	2	16	report timeliness, report integrity, report accuracy, and disposal effectiveness.
51	Components of the military health response capability for public health emergencies index.	3	71	organizational command capability, disease prevention and control capabilities, medical treatment capabilities, and support capabilities.
52	Hospital Coping Ability Evaluation Index System.	2	53	emergency command coordination mechanism, emergency plan, monitoring and early warning capability, laboratory management and diagnosis, information report exchange, emergency personnel, emergency bed, emergency drug reserve, medical treatment measures, and disinfection and purification.
53	A comprehensive assessment tool of the ability of all provinces, autonomous regions (municipalities)/prefectures (states) to respond to public health emergencies.	2	214	command coordination and evaluation, preparation of emergency plans for public health emergencies, training and exercises, risk identification, assessment, and mitigation, monitoring, early warning, epidemiological investigation and response capabilities, laboratory testing, on-site first aid and medical treatment, information reporting, communication, and dissemination, logistics support, public education, and personnel training.
54	Evaluation Index for Emergency Treatment of Sudden Epidemic Event in Luoyang City.	2	14	epidemic situation detection, epidemic response, on-site investigation, on-site treatment, and epidemic event.
55	Henan Province Municipal-level CDC’s assessment system for public health emergencies.	3	66	basic conditions, system construction, monitoring and early warning, on-site disposal, assessment, safeguard measures, and education and training.
56	Grassroots preventive health care center emergency response capacity and evaluation index.	2	29	organization management, technology implementation, resource reserve, monitoring and early warning capability, coordination and cooperation.
57	District-level public health emergency response capability assessment index: (i) The health administrative department evaluation system.	2	54	organizational command, emergency work management system, monitoring and early warning capability, information reporting and release, on-site disposal, emergency team, equipment reserve, training drill, mobilization propaganda, scientific and technological exchanges, and co-operation, recovery, reconstruction, and response assessment.
	(ii) The disease prevention and control institution evaluation system.	2	45	organizational command, emergency work management system, monitoring and early warning, information reporting and release, on-site disposal, laboratory capabilities, equipment reserve, training drills, mobilization publicity, and technology exchange and cooperation.
	(iii) The medical institution evaluation system.	2	23	organizational command, emergency work management system, monitoring and early warning capability, information reporting and release, on-site disposal, equipment reserve, training drills, scientific and technological exchange (and cooperation).
		2	14	organizational command, emergency work management system, information reporting and release, on-site disposal, equipment reserve, and training drills.
58	Emergency capability survey indicator system.	2	38	basic personnel conditions, emergency management mechanism, monitoring and early warning, on-site investigation and handling, laboratory testing, emergency materials reserve, staff training drills, and public information and information channels.
59	The index system of the military’s ability to respond to major natural disasters’ health emergency support capabilities.	3	48	organizational command capability, medical rescue capability, professional strength construction, and service support capability.
60	Competency model of health emergency personnel in county-level disease control institutions.	1	4	personal characteristics, basic knowledge, emergency knowledge concepts, and emergency skills.
61	China’s health sector International Health Regulations (IHR) (2005) analysis of the standard of public health emergency core competency.	2	59	monitoring description, response, risk communication, preparation, infection control, laboratory capabilities, and material and financial support.

**Table 4 ijerph-16-03478-t004:** Characteristics and application of indicators in evaluation system studies.

Institution Category	Most Frequent Indicators	Number of Indicators	Application
State and health administrative department	Management capabilities, improvement capabilities, drill capabilities, reporting capabilities, system construction, monitoring capability, material reserve capability, training capability, early warning capability, social mobilization capability, site disposal capability, risk assessment.	12	District Health Emergency Capability Survey and Evaluation (Shanghai). Township Public Health Emergency Response Assessment (Sichuan Province). Evaluation of the ability of public health emergency response in a province/autonomous region/municipality directly under the central government (National). District-level public health emergency response capacity assessment (Guangdong Province).
Disease prevention and control institutions	Technical expertise, management capabilities, response capabilities, participation capabilities, response capabilities, monitoring capabilities, alert capabilities, response capacity, drill capability, materials reserve support capability, training capability, system construction, cooperation capability, team-building capability.	14	City CDC and District Center for Disease Control and Prevention Health Assessment Capacity Assessment (Shanghai). Evaluation of relative efficiency of health emergency human resource allocation in county and district CDCs (Shandong Province). City and county-level disease prevention and control centers to respond to public health emergencies (Guangxi Zhuang Autonomous Region). Municipal disease prevention and control agency emergency capability assessment (Guangdong Province). Evaluation of infectious disease prevention and control capability of municipal and county-level disease prevention and control centers (Inner Mongolia Autonomous Region). Evaluation of public health emergency response capability in district and township defense offices (Guangdong Province). County-level CDC emergency public health emergency response capability evaluation (national). City district-level disease prevention and control center emergency public health emergency response capability evaluation (Shandong Province). County-level disease prevention and control agency/epidemic station emergency public health incident on-site emergency response capability evaluation (Henan Province). City-level disease prevention and control center emergency public health emergency response capability evaluation (Henan Province). Evaluation of public health emergency response capacity of district and township preventive health care centers (Guangdong Province). Municipal, county-level disease prevention and control center (Guangdong Province).
Medical institutions	Monitoring capability, plan capability, material reserve, training, early warning, management command, information monitoring report, system, personnel, medical technology, education, laboratory, medical treatment, bed, logistics support, communication, on-site disposal, drills.	18	County general hospital and township health center (Guangxi Zhuang Autonomous Region). Top three hospitals (Hebei Province). Hospital (Heilongjiang Province). Basic-level emergency public health emergency response assessment (national). Secondary hospitals (Beijing).
Health supervision agencies	Organizational command, management system, equipment reserve, training drill, response.	5	District Health Emergency Capability Survey and Evaluation (Shanghai). District-level public health emergency response capacity assessment (Guangdong Province).
Schools	Value education, preparation and recovery, detection and monitoring, response, materials learning.	5	No.
Military	Command capability, support capability, equipment and other reserve capability, system construction, response capacity, monitoring capability, medical treatment capacity, early warning capability, education, control capability, service capability, survival capability, team construction, response capability, research capability, drill, rescue capability, mission, medicine reserve.	19	Digital hospitals (all military).
Border port health and quarantine department	Vulnerability, environment, system construction, laboratory construction, network construction, drill, support, on-site disposal measures, monitoring, training, warning.	11	Xinjiang border port.

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
