# Peer review of "Evaluation of Public Health Emergency Management in China: A Systematic Review"

_ijerph, 2019, doi:10.3390/ijerph16183478_

Round 1

Reviewer 1 Report

General overall comments:

1.  The paper can be significantly condensed.  It currently reads as a term paper with all the results of the literature review included as part of the paper. 

2.  There are too many tables.  There should be 2-3 tables summarizing the results of the 61 studies that were included in the final analysis.  The content of the current tables should be condensed and summarized.  The information in Appendix A should be included in the Methods section.

3.  Figure 1 is good and clearly defines studies that were included and excluded. 

Specific comments:

Page 1, Line 30:  Delete "Later, a series" and begin sentence with "More." 

Page 1, Line 31:  Use another term besides "further disease related pressures."

Page 1, Line 38:  Use another word besides "grueling."

Page 1, Lines 40-44: The focus of the paper is stated here, but that does not reflect what is in the results and discussion.

Page 19:  Need to state the overall result of the studies.  The table (pages 20-24) could be deleted.  It is very difficult to read. 

Page 25:  Again, need a longer, more detailed summary of results for Characteristics of Assessment System.

Page 32:  There needs to be more information on what documented results  indicate for China's health emergency management system. 

Page 32:  Sections 3.5.4. and 3.5.5 can be combined.

Page 33:  What are the overall strengths, weaknesses and gaps in the research based on the results of your analysis?

Page 48:  The Conclusions section is very clear and makes the connection to what was found in the results, but there should be more discussion on what the results mean for emergency preparedness in China.  The authors did well in stating steps that can address identified gaps.

Author Response

Responses to the reviewers’ comments:
Reviewer #1:

Revised portion are marked in red in the paper. General overall comments: 

1.  The paper can be significantly condensed.  It currently reads as a term paper with all the results of the literature review included as part of the paper.

Response: Thank you very much for your suggestion. In consideration of your suggestion, we have shortened the paper. 

2.  There are too many tables.  There should be 2-3 tables summarizing the results of the 61 studies that were included in the final analysis.  The content of the current tables should be condensed and summarized.  The information in Appendix A should be included in the Methods section.
Response: Thank you very much for your suggestion. In consideration of your suggestion, we have removed the original Table 1 and Table 2 and extracted the original Table 2 with reference to the literature. We have simplified the original Table 3, simplified the original Table 4 and Table 5, and analyzed and summarized them. The data from Appendix A have been added to the Methods section in the form of a search box. All of this has been done to make the information in the article more concentrated.

Figure 1 is good and clearly defines studies that were included and excluded.

Response: Thank you very much for your comments.

Specific comments:
1. Page 1, Line 30:  Delete "Later, a series" and begin sentence with "More." 

Response: Thank you very much for your suggestion. In consideration of your suggestion, we have deleted “Later, a series” and begin the sentence with “More.”

Page 1, Line 31: Use another term besides "further disease related pressures."

Response: Thank you very much for your suggestion. In consideration of this suggestion, we have modified the text as follows, “had a serious impact on China's politics, economy, and trade, people’s health, and so on.”

Page 1, Line 38: Use another word besides "grueling."

Response: Thank you very much for your suggestion. In consideration of your suggestion, we have used “rigorous” replacement for “grueling.”

Page 1, Lines 40-44: The focus of the paper is stated here, but that does not reflect what is in the results and discussion.

Response: Thank you very much for your suggestion. In consideration of your suggestion, we have modified the text as follows: “There are many types of health emergency management assessments and many methods for health emergency management assessment in China, and the dimensions and indicators of health emergency management assessment are complex. The laws, regulations, and policy basis for health emergency management assessment still need to be further strengthened. The relevance of the whole process of the health emergency management assessment system need to be improved, and the practical application of the health emergency management assessment system still needs to be strengthened.”

Page 19: Need to state the overall result of the studies. The table (pages 20-24) could be deleted.  It is very difficult to read.

Response: Thank you very much for your suggestion. In consideration of your suggestion, we have provided the overall result of the studies, refined the original Table 2, and incorporated the references into Table 2. We have also created a new Table 1.

Page 25: Again, need a longer, more detailed summary of results for Characteristics of Assessment System.

Response: Thank you very much for your suggestion. In consideration of your suggestion, we have reworked this section as follows: “In all the health emergency assessment systems, the maximum number of all level indicators was 214 while the minimum was 4, the maximum number of primary indicators was 15 while the minimum was 2, the maximum number of secondary indicators was 204 while the minimum was 0, and the maximum number of tertiary indicators was 84 while the minimum was 0. The specific results are detailed in Table 2.”

7. Page 32:  There needs to be more information on what documented results indicate for China's health emergency management system.Response: Thank you very much for your suggestion. In consideration of your suggestion, we have reworked this section as follows:

“Indicator Characteristics and Application of Assessment System

               The indicator characteristics and application of the health emergency assessment system included in the study were analyzed according to different institutional categories, which were ‘State and Health Administrative Department,’ ‘Disease Prevention and Control Institutions,’ ‘Health Institution,’ ‘Health Supervision Agency,’ ‘School,’ ‘Military,’ and ‘Border Port Health and Quarantine Department.’”               We have further simplified the original Table 4, combined it with the original Table 5, classified the data according to the main body of the health emergency assessment, and analyzed the application of high frequency indicators and systems to form a new Table 3. We have deleted the original Table 5.

Page 32: Sections 3.5.4. and 3.5.5 can be combined.

Response: Thank you very much for your suggestion. We apologize for our negligence of the term. We have corrected this text as follows: “3.5.4 Health Supervision Agency Assessment Research.”

9. Page 33:  What are the overall strengths, weaknesses and gaps in the research based on the results of your analysis?

Response: Thank you very much for your suggestion. In consideration of your suggestion, we have modified this part of the content as follows: “Through this research, we have gained a comprehensive understanding of the status quo of China’s health emergency assessment and explored the main distribution areas of the representative research of China’s health emergency assessment system, research output, research design, main dimensions, high frequency indicators, major application cities, and health emergencies. The study only included Chinese studies, which affects the comprehensiveness of the research”

10. Page 48:  The Conclusions section is very clear and makes the connection to what was found in the results, but there should be more discussion on what the results mean for emergency preparedness in China.  The authors did well in stating steps that can address identified gaps.

Response: Thank you very much for your suggestion. In consideration of your suggestion, we have modified this part of the content as follows: “The representative research of China’s health emergency assessment system is mainly distributed in the eastern part of China. The design of such research mainly focuses on cross-sectional research, and the dimensions and while high-frequency indicators of the health emergency assessment system are excavated, the application of the health emergency assessment system is applied. Sex has yet to be improved. China should emphasize research and application of health emergency assessment in the central and western regions, including in ethnic minority areas, and the health emergency assessment system should be improved from the perspective of classification and integration of health emergency assessment subjects, as health emergency management assessment is an important part of the health emergency management activities. Through evaluation, problems in health emergency management can eventually be found, and the effective prevention and control of public health emergencies can be achieved to the greatest possible extent. There are many types of and methods for health emergency management assessments, and the related dimensions and indicators are complex. In health emergency management assessment, it is extremely important to select appropriate assessment methods according to different assessment purposes, targets, and contents. The laws, regulations, and policy basis of health emergency management assessment should be strengthened, especially after the establishment of the National Emergency Management Department in April 2018. The most important finding is that the correlation research of the entire process of the health emergency management evaluation system needs to be further explored and improved upon.”

Thank you for your valuable suggestions. We greatly appreciate your time and efforts to improve our manuscript for publication.

Reviewer 2 Report

You have undertaken an important area of study in the field of emergency preparedness, and taken great pains to search the literature and describe your methods.  Unfortunately the details provided on your search and the individual analysis of each included study detracts from the reader getting to your findings and discussion about what dimensions could best be included in a stronger ongoing assessment of the system.  You appear to draw some conclusions about what is needed in the introcudtion (lines 41-44).  The first 2 tables are far too long and detailed to be useful to the reader.  The discussion beginning on line 178 is very informative, and could be accompanied by a useful table.  Table 4 as it now is was not helpful, and it was impossible for this reader to determine what Table 5 was intended to say. 

A very firm editorial hand could re-do a minimum number of tables, develop a box to summarize the method, and make the findings thus more accessible and helpful to the interested reader.

Author Response

Responses to the reviewers’ comments:Reviewer #2:

You have undertaken an important area of study in the field of emergency preparedness, and taken great pains to search the literature and describe your methods.  Unfortunately the details provided on your search and the individual analysis of each included study detracts from the reader getting to your findings and discussion about what dimensions could best be included in a stronger ongoing assessment of the system.  You appear to draw some conclusions about what is needed in the introcudtion (lines 41-44).  The first 2 tables are far too long and detailed to be useful to the reader.  The discussion beginning on line 178 is very informative, and could be accompanied by a useful table.  Table 4 as it now is was not helpful, and it was impossible for this reader to determine what Table 5 was intended to say.

A very firm editorial hand could re-do a minimum number of tables, develop a box to summarize the method, and make the findings thus more accessible and helpful to the interested reader

Response: Thank you very much for your suggestion. In consideration of your suggestion, we have revised and improved the manuscript. Revised portion are marked in red in the paper.

First, we have modified the text (lines 40–43) as follows: “There are many types of health emergency management assessments and many methods for health emergency management assessment in China, and the dimensions and indicators of health emergency management assessment are complex. The laws, regulations, and policy basis for health emergency management assessment still need to be further strengthened.”

Second, we have inductively analyzed the original Table 1 as follows: “In general, the representative research of China’s health emergency assessment system is mainly distributed in the eastern part of China. The most common type of publication is papers in academic journals, and the main type of research is on health emergency response ability. The research design is mostly cross-sectional research, according to health emergency management. The main body classification most frequently includes research on health emergency management evaluation of health administrative departments, disease prevention and control institutions, medical and health institutions, health supervision institutes, schools, military and ports.” We have also deleted the original Table 1, streamlining the article as a whole.

Third, we have stated the overall result of the studies as follows: “Overall, the research quality of China's health emergency management assessment system is relatively good.” We have also refined the original Table 2, incorporated the references into Table 2, and formed a new Table 1.           Fourth, regarding the information starting after line 178, we combined the original Tables 4 and 5 to refine the information. We have reworked this section as follows:

“Indicator Characteristics and Application of Assessment System

The indicator characteristics and application of the health emergency assessment system included in the study were analyzed according to different institutional categories, which were ‘State and Health Administrative Department,’ ‘Disease Prevention and Control Institutions,’ ‘Health Institution,’ ‘Health Supervision Agency,’ ‘School,’ ‘Military,’ and ‘Border Port Health and Quarantine Department.’”

 We have simplified the original Table 4, combined it with the original Table 5, classified the data according to the main body of the health emergency assessment, and analyzed the application of high frequency indicators and systems to form a new Table 4. We have deleted the original Table 5 and formed a new Table 3. The integration of information is more logical and the article is more concise.Fifth, we have improved the overall structure, content, and language of the article, especially at the end of the article, summarizing the research and highlighting the focus of the article. We sincerely hope that through our research, we will make a contribution to the development of China’s health emergency systems as well as global health emergency systems. Thank you for your valuable suggestions. We greatly appreciate your time and efforts to improve our manuscript for publication.

Round 2

Reviewer 1 Report

Lines 30-33:  Sentence needs to be rewritten with correct English structure including not ending a sentence with a preposition.

Line 41:  Define "dimensions" and "indicators."

Lines 43 and 46:  "Further strengthened" is mentioned several times, but the authors do not state HOW the system will be strengthened.

Line 156:  Delete the word "in general" and the sentence is confusing.

Line 159:  Use the word "primarily" instead of "mostly."

Lines 170 - 172:  What is the basis for determining the assessment system is "relatively good?"  If there are cut off points based on the content in Table 1, it should be documented along with the the quantitative score.

Lines 182-186:  Edit and correct grammar and sentence structure.

Table 2 needs to be consolidated.  It is still too long.

Lines 420-423 are confusing.  Consider breaking into two sentences.

Line 429:  Should be "greatest extent possible."

Lines 435-437:  What is stated as a finding is not really a finding.

Author Response

Responses to the reviewers’ comments:
Reviewer #1 (Round 2):

Revised portion are marked in green in the paper. Comments and Suggestions for Authors: 1.  Lines 30-33:  Sentence needs to be rewritten with correct English structure including not ending a sentence with a preposition.
Response: Thank you very much for your suggestion. In consideration of your suggestion, we have modified this part of the content as follows: “More public health emergencies such as the H1N1 flu epidemic in 2009 and the H7N9 avian flu epidemic in 2013 had a serious impact on China’s politics, economy, trade, and people’s health, among others.” 2.  Line 41:  Define "dimensions" and "indicators."
Response: Thank you very much for your suggestion. In consideration of your suggestion, we have modified this part of the content as follows: “For example, the evaluation system named the “Disease control agency health emergency capacity evaluation index” has three dimensions, the system known as the “Evaluation index for public health emergency response capacity of township hospitals” has two dimensions, the system named the “National Health and Family Planning Commission, Health Assessment Capacity Assessment Standard” only has one dimension, the system called “A comprehensive assessment tool of the ability of all provinces, autonomous regions (municipalities)/prefectures (states) to respond to public health emergencies” has 214 indicators, and the evaluation system named the “Competency model of health emergency personnel in county-level disease control institutions” only has four indicators. In general, the dimensions of health emergency assessment are mostly second-level dimensions. The assessment indicators mainly include management capabilities, improvement capabilities, drill capabilities, and reporting capabilities. For purposes of this research, a ‘dimension’ refers to content of public health emergency management features of the institution, such as health emergency response dimension, and ‘indicator’ is used to refer to evaluation features, including emergency command coordination mechanism.” 

Lines 43 and 46:  "Further strengthened" is mentioned several times, but the authors do not state HOW the system will be strengthened.

Response: Thank you very much for your suggestion. In consideration of your suggestion, we have modified this part of the content as follows: “After the establishment of the National Emergency Management Department, it is necessary to pay attention to the differences and connections between the health emergency and national emergency laws and policies. It is necessary to improve the relevance of the entire process of the health emergency management assessment system. The practical application of the health emergency management assessment system also should focus on the application of health supervision agencies, schools, and other institutions.”

4.
Line 156:  Delete the word "in general" and the sentence is confusing.
Response: Thank you very much for your suggestion. In consideration of your suggestion, we have deleted the phrase “in general” and modified this part of the content as follows: “According to the classification of health emergency management subjects most frequently includes research on health emergency management evaluation of health administrative departments, disease prevention and control institutions, medical and health institutions, health supervision institutes, schools, military, and ports.”

5. Line 159:  Use the word "primarily" instead of "mostly."

Response: Thank you very much for your suggestion. In consideration of your suggestion, we have used the word “primarily” instead of “mostly.”

6. Lines 170 - 172:  What is the basis for determining the assessment system is "relatively good?"  If there are cut off points based on the content in Table 1, it should be documented along with the the quantitative score.
Response: Thank you very much for your suggestion. In consideration of your suggestion, we have modified this part of the content as follows: “most studies were given a rating of 2* according to the Ekman quality assessment tool. which score between 17 and 21.”No cut-off points were used based on the content in Table 1.

7. Lines 182-186:  Edit and correct grammar and sentence structure.
Response: Thank you very much for your suggestion. In consideration of your suggestion, we have modified this part of the content as follows: “Overall, in the health emergency assessment systems, the maximum and minimum number of indicators were as follows: all indicators (maximum of 214, minimum of 4); primary indicators (maximum of 15, minimum of 2); secondary indicators (maximum of 204, minimum of 0); tertiary indicators (maximum of 84, minimum of 0). The specific results are detailed in Table 2.”

8.Table 2 needs to be consolidated.  It is still too long.
Response: Thank you very much for your suggestion. In consideration of your suggestion and based on the contents of Table 2, we have consolidated this table, keeping the more important parts.

9.Lines 420-423 are confusing.  Consider breaking into two sentences.
Response: Thank you very much for your suggestion. In consideration of your suggestion, we have modified this part of the content as follows: “The design of such research mainly focuses on cross-sectional studies. While the dimensions and high-frequency indicators of the health emergency assessment system have been extracted, the applicability of the health emergency assessment system needs to be improved.”

10. Line 429:  Should be "greatest extent possible."
Response: Thank you very much for your suggestion. In consideration of your suggestion, we have modified this to “greatest extent possible.”

11.
Lines 435-437:  What is stated as a finding is not really a finding.
Response: Thank you very much for your suggestion. In consideration of your suggestion, we have modified this part of the content as follows: “…there are many types of health emergency management assessments and multiple health emergency management assessment methods in China. The dimensions and indicators of health emergency management assessment are also complicated. In general, the dimensions of health emergency assessment are mostly in the secondary dimension. The evaluation indicators mainly include management ability, improvement ability, exercise ability, and reporting ability. The laws, regulations and policy basis for health emergency management assessment also need to attend to the differences and linkages between national health emergencies and national emergency laws, regulations, and policies. It is necessary to improve the relevance of the entire process of the health emergency management assessment system. The practical application of the health emergency management assessment system also needs to pay attention to the application of health supervision, schools, and other institutions. The correlation research of the entire process of the health emergency management evaluation system needs to be further explored and improved. In addition, China should build a health emergency management assessment system that is in line with international standards and focuses on Chinese characteristics, which needs further exploration in the future.”

Thank you for your valuable suggestions. We greatly appreciate your time and efforts for improving our manuscript for publication.

Reviewer 2 Report

Thank you for taking the initial review comments seriously and making significant changes in the manuscript. It is now much more readable and the analysis and findings more useful to the reader.  

Author Response

Responses to the reviewers’ comments:Reviewer #2 (Round 2):

Thank you for taking the initial review comments seriously and making significant changes in the manuscript. It is now much more readable and the analysis and findings more useful to the reader.

Response: Thank you very much for your advice, guidance, and affirmation. In the future, we will continue to work hard and strive for more scientific achievement.
